# A Soft Robot Driven by a Spring-Rolling Dielectric Elastomer Actuator with Two Bristles

**DOI:** 10.3390/mi14030618

**Published:** 2023-03-08

**Authors:** Yangyang Du, Xiaojun Wu, Jiasheng Xue, Xingyu Chen, Chongjing Cao, Xing Gao

**Affiliations:** 1School of Mechanical and Electrical Engineering, Xi’an University of Architecture and Technology, Xi’an 710049, China; 2Research Centre for Medical Robotics and Minimally Invasive Surgical Devices, Shenzhen Institutes of Advanced Technology (SIAT), Chinese Academy of Sciences, Shenzhen 518055, China

**Keywords:** SRDE, soft robotics, bristles, crawling, resonance

## Abstract

Confined space searches such as pipeline inspections are widely demanded in various scenarios, where lightweight soft robots with inherent compliance to adapt to unstructured environments exhibit good potential. We proposed a tubular soft robot with a simple structure of a spring-rolled dielectric elastomer (SRDE) and compliant passive bristles. Due to the compliance of the bristles, the proposed robots can work in pipelines with inner diameters both larger and smaller than the one of the bristles. Firstly, the nonlinear dynamic behaviors of the SRDE were investigated experimentally. Then, we fabricated the proposed robot with a bristle diameter of 19 mm and then studied its performance in pipelines on the ground with inner diameters of 18 mm and 20 mm. When the pipeline’s inner diameter was less than the outer diameter of the bristles, the bristles remained in the state of bending and the robot locomotion is mainly due to anisotropic friction (1.88 and 0.88 body lengths per second horizontally and vertically, respectively, in inner diameter of 18 mm and 0.06 body length per second in that of 16 mm). In the case of the pipeline with the larger inner diameter, the bristles were not fully constrained, and a small bending moment applied on the lower bristle legs contributed to the robot’s locomotion, leading to a high velocity (2.78 body lengths per second in 20 mm diameter acrylic pipe). In addition, the robot can work in varying geometries, such as curving pipes (curve radius ranges from 0.11 m to 0.31 m) at around two body lengths per second horizontally and on the ground at 3.52 body lengths per second, showing promise for pipeline or narrow space inspections.

## 1. Introduction

Soft robots have good structural flexibility and compliance, which can significantly reduce the robot’s structural and control complexity. Additionally, compared with the conventional rigid robots, it is easier to achieve a breakthrough in miniaturization and good adaptability in uncertain environments. Therefore, soft robots for industrial confined space searches, such as pipeline inspections, have attracted lots of attention [1,2]. A variety of bionic structure robots, such as the bionic worm [3,4], inchworm [5,6], snake [7,8], starfish [9,10], and jellyfish [11,12], have been designed to adapt to different uncertainties. In addition, based on smart materials and their structures, many soft robots have special functions, such as jumping and hopping [13,14], climbing walls [15], peristaltic in sub-centimeter gaps [16], climbing in pipes horizontally or vertically [17,18], passing through bend pipes [19,20], flying [21], and rolling [22,23].

Compared with the shape memory alloy and pneumatic actuation, dielectric elastomers (DEs) have the advantages of fast response, easy control, and high energy efficiency [12,24,25,26,27,28,29,30]. Moreover, DEs exhibit a higher efficiency of electromechanical conversion than that in actuators driven by piezoelectric ceramics and magnetic fields [1,31]. Thereby, many robots driven by DEs have been proposed.

Similar to the motion mechanisms adopted by most soft robots, the robots driven by DEs mainly achieve directional motion through an anchoring structure, such as electrostatic adsorption [15,32] and mechanical anchoring [20], anisotropic structure [5,13,33,34,35] (e.g., bristles), or legged motion structure [36]. However, most of the current dielectric elastomer robots (DERs) tends to work in a single environment; for example, the robot in references [5,33,36,37,38] can only crawl on the ground, while the robot with two complex anchors in reference [20] can only adapt to pipelines whose inner diameter is less than the anchors external contour. To solve this problem, inspired by the driving principle of bristle robots, we designed a crawling robot driven by SRDE with wide adaptability. It can not only crawl in pipelines with an inner diameter larger than the external contour of bristle, especially on the ground horizontally (faster than two body lengths per second), but also move in pipelines with an inner diameter smaller than the external diameter of bristle in a creeping motion horizontally (close to two body lengths per second) and vertically (close to one body length per second). In addition, this bristle has a simple design and an easy fabrication process. Hence, it exhibits high application adaptability in a variety of environments.

The remaining paper is structured as follows. In Section 2, the structures and working principles of SRDE and the corresponding soft robot are introduced. Subsequently, the fabrication process is described. In Section 3, the dynamic response of SRDE is tested. In Section 4, the friction of the bristle used in the proposed robot is measured. In Section 5, the horizontal crawling performance of the robot in different-diameter acrylic tubes are tested. Furthermore, the proposed robot is proven to have the additional abilities to crawl in vertical acrylic tubes, curved pipes, and on horizontal ground. The final part is the discussion and conclusion.

## 2. Design and Fabrication of SRDE and the Robot

### 2.1. Design and Principle of SRDE and the Robot

The proposed SRDE shown in Figure 1 consists of a rolling DE (RDE), a coupled spring, double endcaps, copper foils, and wooden sticks at both ends axially. When the bristles are structured at the two ends, the developed SRDE is able to crawl in pipelines and on the ground.

In the assembly process of SRDE, the spring between two endcaps remains in compression. After assembly and staying free, the inside RDE transforms to a stretched state due to partial release of the compressive spring, as shown in Figure 2a. When subjected to a high voltage, the RDE becomes compressed radially for the Maxwell stress generated by the electric field, causing the spring to further release, as shown in Figure 2b. If using an AC signal instead of the constant voltage, the force *F_D_* applied to the endcap by RDE shown in Figure 2c will change in the period correspondingly, causing the axially periodic elongation and contraction of the SRDE.

When the pipe inner diameter is less than the outer diameter of both end bristles, the periodic motion of the SRDE can make the robot achieve a directional locomotion shown in Figure 2d–f for the existed anisotropic friction [39]. During the elongation of the SRDE (Figure 2d transiting to Figure 2e), the front and rear bristles move forward (*dx*_2_) and backward (*dx*_1_), respectively. However, the backward friction is greater than forward friction (*F_fb_* > *F_ff_*), resulting in *dx*_1_ < *dx*_2_. Similarly, when the SRDE shrinks (the transformation from Figure 2e,f), there exists a relationship *dx*_4_ < *dx*_3_. Finally, in one cycle, the center of robot moves forward for a displacement (*dx*_3_ + *dx*_2_
*− dx*_1_
*− dx*_4_)/2. If we assume that both bristles have the same *F_fb_* and *F_ff_* (although it is difficult to realize), the periodic movement will equal to *dx*_3_ *− dx*_1_ approximatively.

If the pipe inner diameter exceeds the bristles’ outer diameter, there exists another way to realize directional crawling, as shown in Figure 3 [13,40]. Considering that both the end-bristle lower legs are in contact with the pipe wall in the same slope, during the elongation of the SRDE (Figure 3a), since the bristle upper legs are free of external force, a counterclockwise bending moment *M_fe_* and a clockwise bending moment *M_re_* will apply to the front-bristle lower leg and the rear-bristle lower leg, respectively. These two bending moments make the front-bristle lower-leg tip trend to move up and left and the rear-bristle lower-leg tip trend to move down and right, leading to a decreasing friction *F_ff_*, and an increasing friction *F_fb_*. Therefore, the forward displacement *dx*_2_ of the front bristle is higher than the backward displacement of the rear bristle *dx*_1_. Similarly, during the shrinkage of the SRDE (Figure 3b), the forward displacement of the rear bristle is higher than the backward displacement of the front bristle. Finally, after an entire period, the middle body crawls forward for a displacement *dx*_2_ – *dx*_1_ (here we also assume that both bristles have the same nominal *F_ff_* and *F_fb_*).

### 2.2. The Fabrication Process

The fabrication of the robot is divided into three steps: RDE preparation, SRDE assembly, and robot assembly.

(A) The RDE preparation. This process is similar to reference [41], where we select the silicone elastomer (Elastosil 2030, thickness 100, Wacker Chemie AG, Munich, Germany) whose dielectric strength is 80–100 kV/μm as the DE membrane. Moreover, in order to reduce the cost and prolong the working life, we proposed a new compliant electrode comprised of Smooth-on Ecoflex 0030, RT625 (Elastosil, Wacker AG), carbon black powder, and isopropyl alcohol, as shown in Table 1. These proportional substances are mixed by a centrifugal mixer and then scraped on a PET sheet to form a film with a thickness of 0.025 mm by a cast film machine. After being cured, the compliant electrode is cut into the symmetrical pattern by a laser cutting machine. Subsequently, referring to [41], the pasting and rolling process is completed, and the RDE is shown in Figure 4a,b.

(B) The assembly of the SRDE. Firstly, one end of the RDE is attached to the 3D-printed endcap, and then both of them are placed on the auxiliary positioning device shown in Figure 4b and are secured with screws and nuts. Then, the spring is housed on the endcap and bonded together with liquid adhesive. Similarly, the other endcap is fixed on the device subsequently, and liquid adhesive is used to bond it and the spring. In order to reduce weight as well as facilitate the testing and robot assembly, the fasteners are re-loosened after the SRDE is assembled. Then, the liquid adhesive is also used to bond the two hemi-endcaps together. Note that the excess of one endcap should be cut during the dynamic test, while that of both endcaps should be removed before the robot assembly. It should be pointed out that in this work, the stiffness of the spring is 0.35 N/mm, and its initial height is 40 mm. When assembling the spring, the height is compressed down to 35 mm. After assembly, the spring extends to 37.3 mm, and the body length of the SRDE is about 52 mm.

(C) The assembly of the robot. As shown in Figure 1, the robot consists of the SRDE in the middle space and two bristles at both ends. The bristle is made from 0.2 mm PET film in two steps: (1) The designed pattern is cut by a laser cutter, which then is manually stamped by a 3D-printed die shown in Figure 4c. The angle between the bristle legs and the tube wall is 63° ± 2° (measured in photograph). (2) In the center of the bristle, one 1 mm thick acrylic ring is glued to improve the body rigidity. Finally, two bristles are housed on both ends of the SRDE through the axial stick with liquid adhesive. The external diameter of the robot is 19 mm, the length is about 60 mm, and the mass is 3.8 g.

In addition, the RC time can affect the dynamic response of DEA [42]. In this work, the initial capacitance of the SRDE is about 400 ± 30 pF and the electrode resistance is about 395 ± 50 kΩ, so the RC time is about 0.16 ms. Hence, when the driving frequency is less than 1000 Hz, the effect of electrical response can be ignored [42,43].

## 3. Dynamic Characterization of the SRDE

The kinematics of the crawling robot are related to the dynamics of the SRDE. In order to analyze the dynamic response, we designed a dynamic test device, as shown in Figure 5. The SRDE is housed on the 3D-printed bracket through the lower endcap by screws and nuts. The upper stick is limited in the axial direction by the axle hole at the upper end of the bracket and coated with vegetable oil to reduce frictional influence.

Following references [43,44], we use the “chirp” function in MATLAB^®^ to generate sweep frequency signals, which are then converted to analog signals and transmitted into a high voltage amplifier (HVA, 10/40A-HS, Trek) by Data Acquisition (DAQ, NI-USB 6363, National Instruments). Finally, a high voltage with dynamic frequency sweeping excitation is applied to the SRDE. Figure 6a shows the schematic diagram of the sweep bias signal, indicating that the AC signal frequency continuously changes from 0 to 4 Hz at a rate of 1 Hz/s with the peak voltage (*U_p_*) of 4 kV. Figure 6b shows the dynamic response results of the SRDE sweep frequency from 0 to 450 Hz at different *U_p_* values. It can be seen that the SRDE exhibits a significant amplitude only when the excitation frequency reaches the resonant frequency. As *U_p_* increases, the amplitude increases obviously. According to the experimental results, as *U_p_* increases to 4.5 kV, there are three peaks, whose corresponding frequencies are 65 Hz, 130 Hz, and 367 Hz, respectively. The middle peak has the maximum amplitude of 2.73 mm.

The “chirp” function in MATLAB^®^ can analyze the driver dynamics quickly by changing the excited frequency constantly. However, the subsequent crawling robot is tested at a fixed frequency. Thus, 1 Hz, 20 Hz, 40 Hz, 65 Hz, 130 Hz, and 367 Hz were selected for fixed frequency measurements to observe the displacement response. The test results under the *U_p_* of 4 kV are shown in Figure 7, and the amplitude is the sum of the absolute values of the maximum and minimum displacement over a period. It can be seen that the amplitudes of 1 Hz, 20 Hz, and 40 Hz are very close to 0.18 mm. The amplitude at 65 Hz is higher than 0.18 mm, but its displacement curve is different from that of the other frequencies, similar to the dynamic response at 50 Hz in reference [20]. The amplitude is the highest at 130 Hz, reaching 1.72 mm. The amplitude at 367 Hz is 0.24 mm, slightly higher than that of 1 Hz, 20 Hz, and 40 Hz. It should be pointed out that a significant phase delay between the displacement and the signal appears when the excited frequency exceeds 65 Hz. In addition, the phase delay increases with increases in the excited frequency, which is related to the modulus and viscosity of the DE material [20,43].

## 4. The Friction Test of Bristles

The geometry of the bristles is shown in Figure 8b. The tips of the bristle legs are triangular, and the width is 1.5 mm, so it is also called the “Triangle 1.5 Bristle”. Considering that the external contour of the SRDE is 13 mm, the outer diameter of this bristle is designed to be 20 mm before being deformed. After being stamped by the die, as shown in Figure 4c, the bristle can form a naturally inclined angle (63° ± 2°) along with the pipe wall, and accordingly, the outer diameter becomes about 19 mm.

Because the outer diameter of the bristles is 19 mm, we set the robot crawling in the pipe with an inner diameter of 18 mm as the initial target. Thereby, the friction force is tested in the acrylic pipe with the inner diameter of 18 mm first, and the test device is shown in Figure 8a. Moreover, there is a laser displacement sensor located at the end of the device to measure the movement of the robot, which is not shown Figure 8a. Similar to the dynamic response test of the SRDE, the load cell signal is input to the DAQ and then read by the MATLAB^®^. When testing, the linear motor pulls the robot in one direction by a fishing line at a rate of 1 mm/s.

Figure 8c,d shows the friction measurement results, indicating that this bristle has significant anisotropy friction; the static friction force and dynamic friction force of moving forward are significantly lower than those of moving backward. When moving forward, the static friction and dynamic friction forces are 0.23 N and 0.13 N, and when moving backward, the static friction and dynamic friction forces are 0.12 N and 0.07 N.

## 5. Performance Test of the Crawling Robot

### 5.1. In Horizontal Acrylic Pipe with 18 mm Inner Diameter 

To prove the robot with Triangle 1.5 bristles is capable of crawling horizontally in an acrylic pipe with an 18 mm (Φ18) inner diameter, the crawling rate at different frequencies is tested under a *U_p_* of 4 kV, and the results are shown in Figure 9 and Figure 10.

Figure 9 shows the crawling locomotion of the robot in one period under the excited frequency of 1 Hz. The displacement data are measured with double-laser displacement sensors respectively located at both ends of the pipeline, as shown in Figure 9a. From Figure 9b, it can be seen that the crawling locomotion in one period consists of four steps: A, B, C, and D. In step A (0 to *U_C_* (~2.7 kV), also called critical voltage), the front bristle, rear bristle, body length, and center of mass have little displacement because of the lower driving force produced by the voltage-induced Maxwell stress. In step B, the input voltage further increases to *U_p_*, resulting in a higher driving force and the front bristle moving forward obviously. However, because the friction *F_fb_* is much higher than *F_ff_*, it is hard to see that the rear has significant backward movement in this stage. Similarly, in step C, the rear bristle also cannot move back visibly. In addition, due to the inertia, when the voltage decreases from *U_p_* to *U_C_*, the front bristle moves forward for a little displacement that is significantly lower than that in step B. In step D, as the voltage further decreases, the Maxwell stress drops significantly, leading to the body shrinking and the rear bristle relatively moving forward for a large displacement. Additionally, this result proves the principle shown in Figure 2.

Figure 10 shows the influence of the excited frequency on the crawling performance. In the range of 1~50 Hz, the robot crawls slowly, less than 4 mm/s, where a positive linear relationship exists between the speed and the excited frequency. Figure 9b shows that the backward displacements of both end bristles are little, so there exists:(1)$$v_{c}\propto \frac{2\pi }{\omega }\cdot \Delta b$$
where *v_c_* is the crawling speed, *ω* is the circular frequency of the excited signal, and ∆*b* is the body length variation in one period. At low frequencies, the amplitude of the SRDE is almost independent of frequency (as shown in Figure 7, the amplitudes of 1 Hz, 20 Hz, and 40 Hz are about 0.18 mm), resulting in the ∆*b* of the robot body in the crawling process being frequency-independent (as shown in Figure 9, about 0.08 mm). Hence, the crawling rate increases linearly with the rise in driving frequency.

As the driving frequency further increases, the crawling rate reaches the first peak of 10.3 mm/s at 110 Hz. In addition, at 190 Hz, the robot’s crawling speed reaches the maximum of 32.4 mm/s, about 0.54 BL/s (body lengths/second). Further increasing the excited frequency, the crawling speed drops evidently. These results are related to the resonant response of the SRDE, which is the driving unit of the robot.

However, note that the resonant frequency of the robot is much higher than that of the SRDE with 130 Hz. The reason is as following: the SRDE is a single-degree-of-freedom vibration system, while the robot is a dual-degree-of-freedom vibration system, as shown in Figure 11.

Herein, we simplify the SRDE to a dynamic system consisting of a linear spring, a linear damping, and a vibrator, so its natural frequency *f_s_* refers to:(2)$$f_{s}=\frac{1}{2\pi }\sqrt{\frac{k_{1}}{M_{ES}}}$$
where *k*_1_ and *M_ES_* are the equivalent stiffness and the equivalent mass, respectively. The robot can be simplified as a dynamic system of a linear spring and a linear damping sandwiched between two vibrators with same mass. Because the constant force (referring to the friction between the robot and environment) has no influence on the resonant frequency, the natural frequency *f_R_* can be described as:(3)$$f_{R}=\frac{1}{\pi }\sqrt{\frac{k_{1}}{M_{ER}}}$$
where *M_ER_* is the equivalent mass of the robot. Combining the above two equations, we can obtain:(4)$$f_{R}=2f_{s}\sqrt{\frac{M_{ES}}{M_{ER}}}$$

According to the mass test and Rayleigh’s method, *M_ER_* and *M_ES_* are 3 g and 1.6 g, respectively. Hence, *f_R_* equals 190 Hz.

Figure 6 indicates that *U_p_* has a positive effect on the amplitude of the SRDE, implying that varied *U_p_* can cause a different crawling performance, as illustrated in Figure 12. When *U_p_* is lower than *U_C_* (~2.7 kV) in the range of 1 kV to 2.5 kV, the robot has no ability to crawl. As *U_p_* increases to 3 kV, the robot begins to crawl slowly at a maximum rate of 1.53 mm/s. When *U_p_* reaches 4.7 kV, the maximum speed reaches 113.2 mm/s, equivalent to 1.88 BL/s, as shown in Figure 13 and Appendix A. In addition, the results show that *U_p_* also affects the resonant frequency of the robot: from 3.0 kV to 4.0 kV, the resonant frequency gradually decreases from 202 Hz to 187 Hz. However, as the *U_p_* further increases, the resonant frequency remains constant. It should be pointed out that the resonant frequency shown in Figure 10 has a little difference from that shown in Figure 12. This is because the test frequency increments in Figure 10 are 5 Hz or 10 Hz, larger than the increments of 2 Hz or 3 Hz in Figure 12.

### 5.2. In Horizontal Acrylic Pipe with 20 mm Inner Diameter

Firstly, the friction of the robot crawling in an acrylic pipe with an inner diameter of 20 mm (Φ20) is tested by the same method as shown in Figure 9a. The result is shown in Figure 14, where we can hardly observe significant differences between forward motion and backward motion. However, note that the legs of the front bristle and rear bristle are in parallel and at an acute angle to the tube wall. This results in the crawling mode shown in Figure 3, and Figure 15 demonstrates the crawling capability.

Similar to Figure 10, Figure 15 shows the crawling performance under different excited frequencies with a *U_p_* of 4 kV. As shown, the robot moves little in the low frequencies (≤40 Hz). As the frequency increases, a peak velocity of 11 mm/s appears at 100 Hz. Then, when the frequency increases to 190 Hz, the velocity reaches a maximum of 167 mm/s, equivalent to 2.78 BL/s, as shown in Figure 16 and Appendix A. As the excited frequency further increases, the crawling rate descends and rises subsequently in the range of 210 Hz to 220 Hz, which is the same phenomenon as in the previous crawling performance in the Φ18 pipe. The reason for this phenomenon needs to be studied in the future. Moreover, the above result indicates that when the inner diameter of the pipe is larger than the external contour of the robot, the robot can only crawl visibly when the excited frequency is higher than 40 Hz. The reason may be related to the amplitude and frequency, which also need to be further studied in the future.

Similar to the crawling performance in the Φ18 pipe, *U_p_* also affects the crawling performance in the Φ20 acrylic pipe. The influence of *U_p_* is shown in Figure 17. When *U_p_* is lower than 2.5 kV, the speed increases slowly. However, as *U_p_* increases above 2.5 kV, the speed increases significantly. In addition, the influence of *U_p_* on the resonant frequency in Figure 17 is similar to that in Figure 12: when *U_p_* is lower than 2.5 kV, resonant frequency decreases with the increase of voltage; as *U_p_* exceeds above 2.5 kV, the resonance happens at a constant frequency of 190 Hz, which is slightly higher than the 187 Hz of the crawling in the Φ18 pipe.

### 5.3. Further Study in Other Environments

Figure 8c shows that the static friction difference between the Triangle 1.5 bristles moving forward and backward is above 0.1 N, greater than the gravity of the robot (0.037 N). Thus, we hypothesize that this bristle can achieve vertical crawling in the Φ18 acrylic pipe, which is proven in Figure 18a and Appendix A. At the *U_p_* of 5.5 kV and the excited frequency of 188 Hz, the robot’s crawling rate is about 51 mm/s, equivalent to 0.85 BL/s. This speed is much lower than the performance in the horizontal Φ18 pipe, which is related to the negative influence of the gravity and the little dynamic friction difference (~0.054 N).

The robot body is designed with a linear spring and soft rolling DE, and thus, it could be bent passively to adapt to the curved pipe. Accordingly, we test the crawling ability of the robot inside the oil-coated curved pipe to simulate environments in the industry of curved pipes with oil, as shown in Figure 18b,c and Appendix A. In the Φ20 PU spring tube with a bending radius of 313.5 mm, the crawling speed can reach 122 mm/s at *U_p_* of 5 kV and an actuated frequency of 186 Hz. In the bending radius of the 110 mm tube, at 5.5 kV and 186 Hz, the average speed can reach 116.7 mm/s. These two results show that the decrease of the pipe bending radius has a negative effect on the crawling speed, similar to the results in the reference [20].

Based on the fact that the robot can crawl in the acrylic pipe whose inner diameter is larger than the robot’s external contour, we speculate that the robot can also crawl on a flat surface, which is demonstrated in Figure 18d. In case of *U_p_* being 4.5 kV and a driving frequency of 186 Hz, the crawling speed can reach 211.4 mm/s, equivalent to 3.52 BL/s.

The above results reveal that the peak voltage has a positive effect on the crawling performance. However, it should be pointed out that in order to protect the devices and robots, the *U_p_* in all above experiments must not exceed 5.5 kV.

## 6. Discussion and Conclusions

In this work, we proposed an in-pipe crawling robot that can adapt to two conditions. In the case that the bristles’ outer diameter is larger than the inner diameter of the pipe, the bristle legs bend due to the antagonistic environment, forming the anisotropy of friction. In addition, in this condition, the robot crawls in horizontal and vertical tubes in peristaltic mode.

When the outer diameter of the bristles is smaller than the inner diameter of the pipe, the robot can also crawl because of the function of the bending moments produced by the body length variation, which is also suitable for crawling on the ground. Furthermore, the crawling rate is significantly higher than that of crawling in the pipe with the inner diameter smaller than the bristle outer contour due to the much smaller friction of the former.

The shape of robot bristles is a circular uniform layout; thus, the robot has an anti-interference ability for ground crawling (i.e., crawling after turning over). Therefore, the robot can be applied to search and rescue operations in confined spaces and other similar situations. The ability to crawl on the ground also reveals that this mode allows the robot to still crawl horizontally in a pipe whose inner diameter is much greater than the bristles’ outer diameter with a fast speed. In addition, due to the soft structure, the driving unit has a certain compliance flexibility, allowing the robot to also crawl in the bending pipe.

Compared with other crawling robots driven by DEs [15,20,32,33,34,35,36], the characteristics mentioned above give our proposed robot the ability to work in pipeline detection and restricted space searches simultaneously. Because the DE, compliant electrodes, springs, endcaps, and bristles are all fabricated with inexpensive commercial materials, the robot is low-cost. Moreover, our robot exhibits a long working life based on the fact that the robot for the above test is still functional. In particular, the bristles are cut by lightweight PET sheet and prepared by manual mold, are easy to carry, and are able to adjust the outline size according to the environment. Herein, this proposed robot can promote the application of soft robots driven by DEs in industrial inspection.

However, there are still some problems with our robot to be solved. In peristaltic crawling, the inner diameter of the pipe has a great influence on the peristaltic speed. In the later stage, it is necessary to develop bristles with variable profile sizes or bristles with an adjustable number of legs. Moreover, at present, the robot crawls passively, adapted to the bend radius, leading to a large critical turning radius. In the later work, the external spring will be used to support adding DEs with distributed compliant electrodes to have active bending functions.

## Figures and Tables

**Figure 1 micromachines-14-00618-f001:**
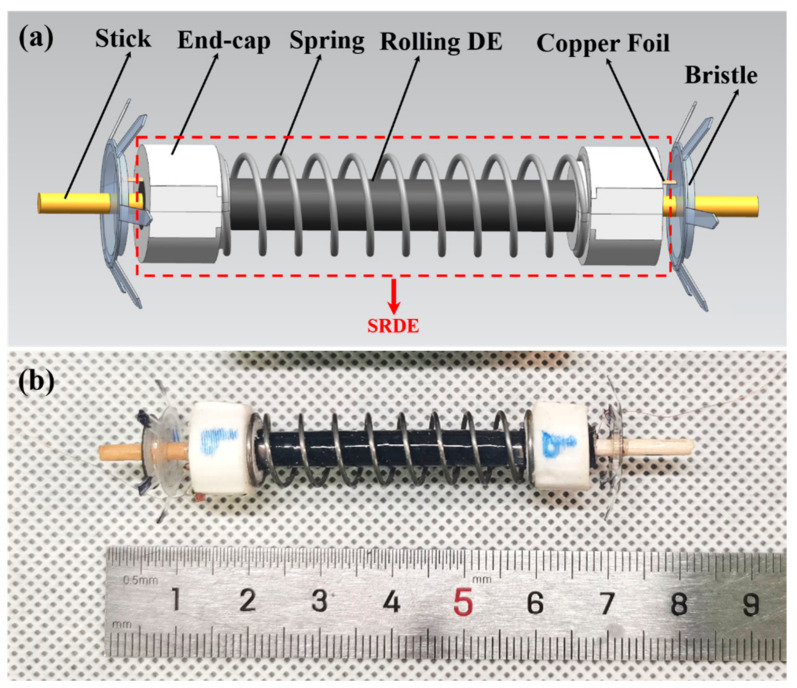
(**a**) Schematic diagram of the crawling robot. (**b**) The physical crawling robot.

**Figure 2 micromachines-14-00618-f002:**
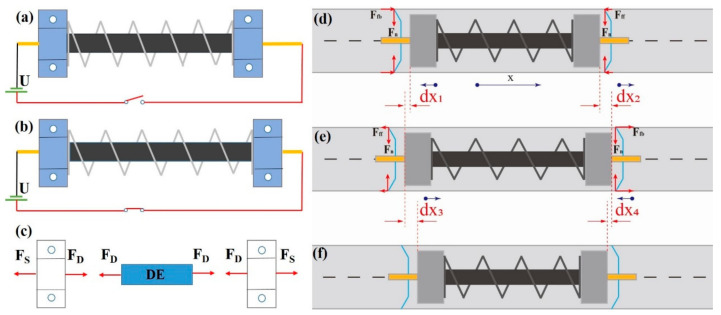
The driving principle of the SRDE (**a**–**c**) and the crawling law of the robot (**d**,**e**). (**a**) The schematic diagram of the SRDE with no voltage. (**b**) The schematic diagram of the SRDE under high voltage. (**c**) The force analysis of SRDE. (**d**) The external force of the robot from contraction to elongation. (**e**) The external force of the robot from elongation to contraction. (**f**) The position of the robot after one period.

**Figure 3 micromachines-14-00618-f003:**
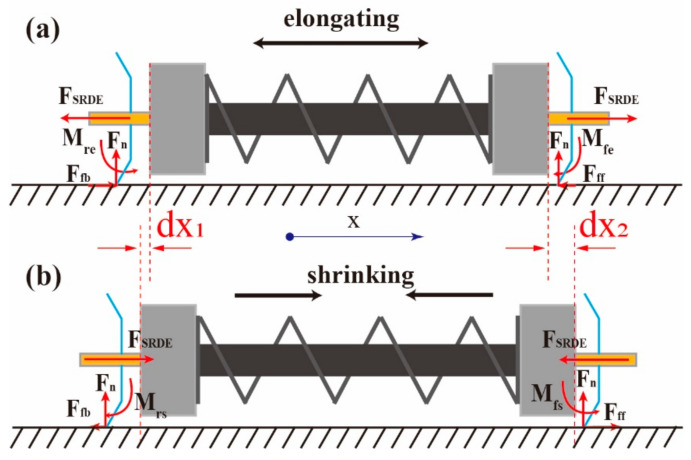
The crawling principle of the robot in a tube with an inner diameter higher than outer diameter of bristle. (**a**) The external force of the bristle-leg tip from contraction to elongation. (**b**) The external force of the bristle-leg tip from elongation to contraction.

**Figure 4 micromachines-14-00618-f004:**
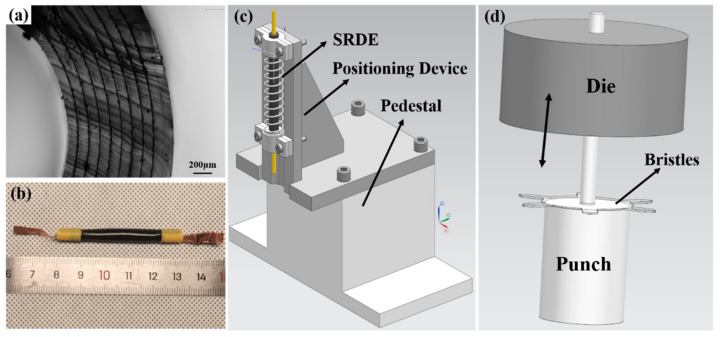
(**a**) The physical body of the RDE (**b**) The cross-section of the RDE. (**c**) The assembly device of the SRDE. (**d**) The stamping die of the bristle (each bristle is stamped five times manually in this work).

**Figure 5 micromachines-14-00618-f005:**
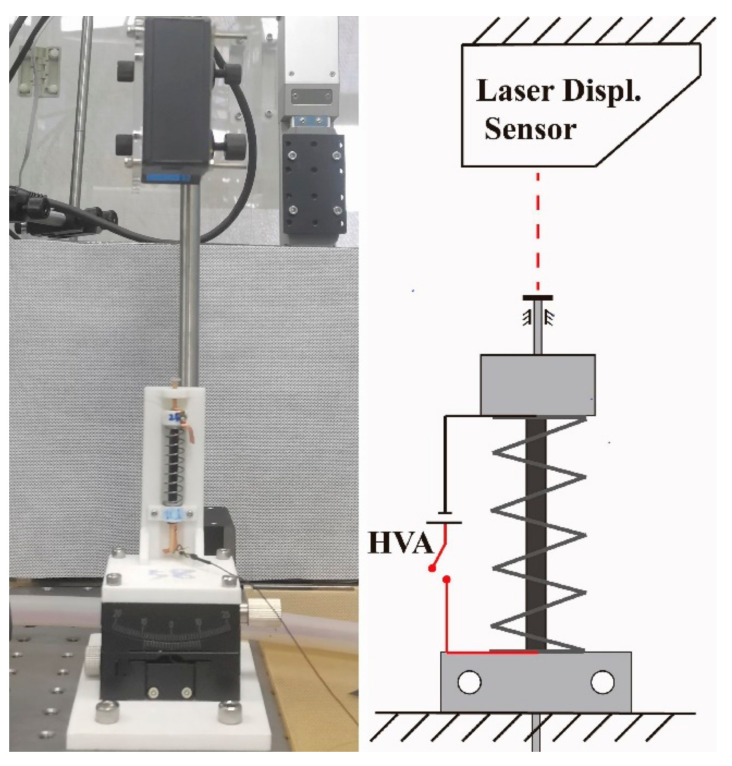
Photo and schematic diagram of the experimental setup for dynamic response tests.

**Figure 6 micromachines-14-00618-f006:**
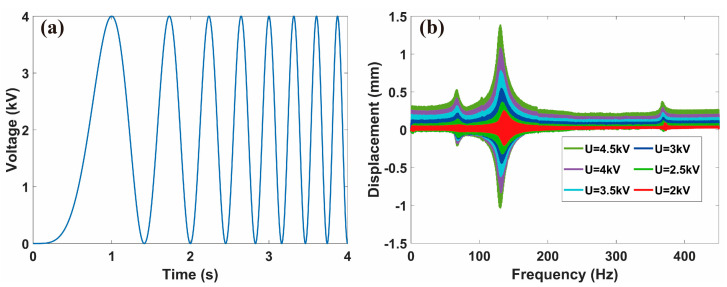
(**a**) Example of a sinusoidal frequency sweep with a DC bias voltage. (**b**) The experimental results of the SRDE in frequency sweep test (sweep from 0 to 450 Hz at a rate of 1 Hz/s) under peak voltage of 4 kV.

**Figure 7 micromachines-14-00618-f007:**
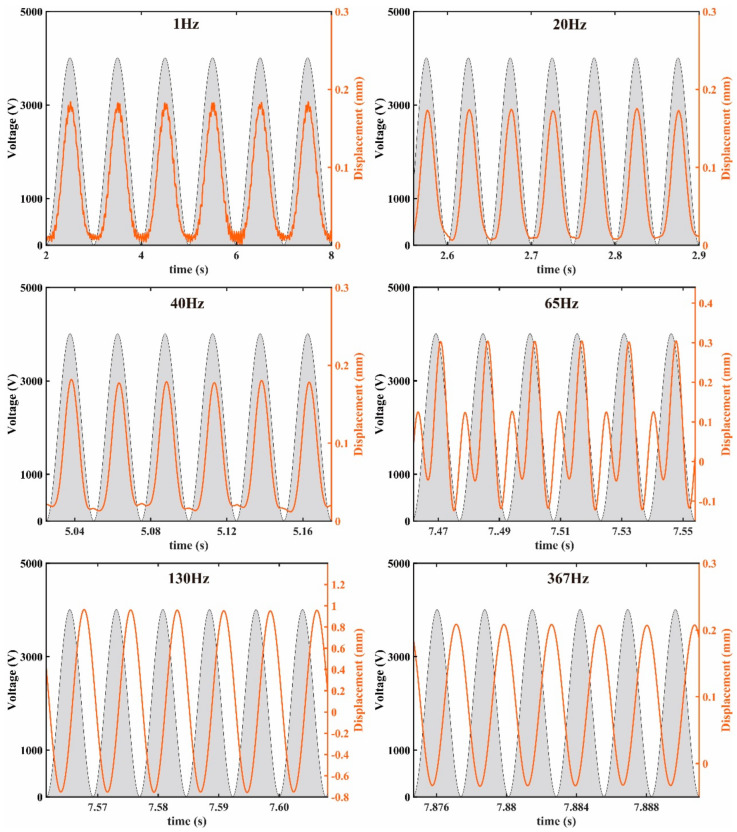
The experimental dynamic response of the SRDE with different fixed frequencies.

**Figure 8 micromachines-14-00618-f008:**
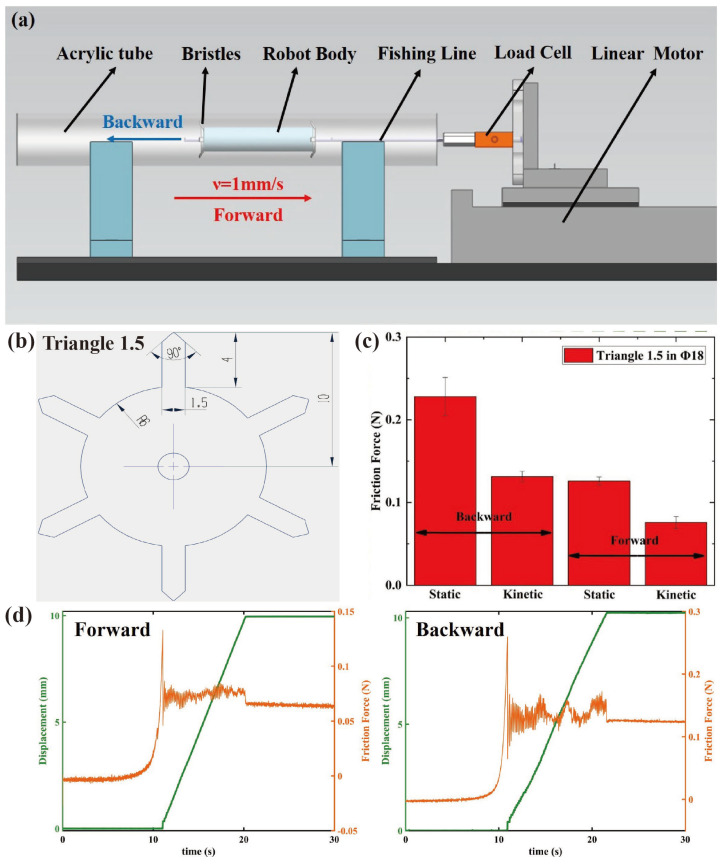
(**a**) The schematic diagram of the bristle friction test device. (**b**) The geometry of bristle. (**c**) The friction test results of the bristle in 18 mm diameter acrylic pipe (tested five times and the results are averaged). (**d**) The friction test data of moving forward and backward.

**Figure 9 micromachines-14-00618-f009:**
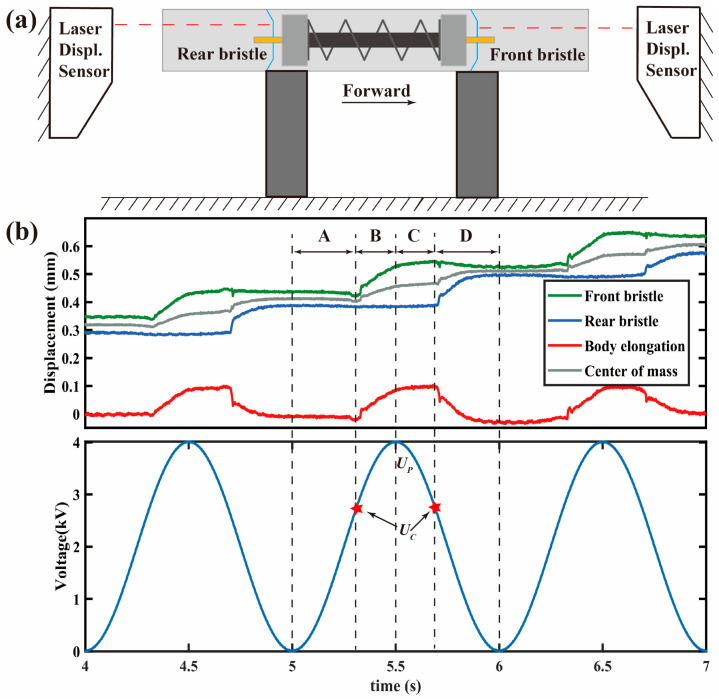
(**a**) The schematic diagram of displacement test with laser displacement sensors; (**b**) Locomotion of the robot crawling in Φ18 acrylic pipe during one period; the displacement signal of front bristle and rear bristle and voltage signal is real-time, which were obtained from the DAQ; the body elongation here refers to its variation, which equals to displacement difference between front bristle and rear bristle; the displacement of center of mass is sum of the displacements of those two bristles.

**Figure 10 micromachines-14-00618-f010:**
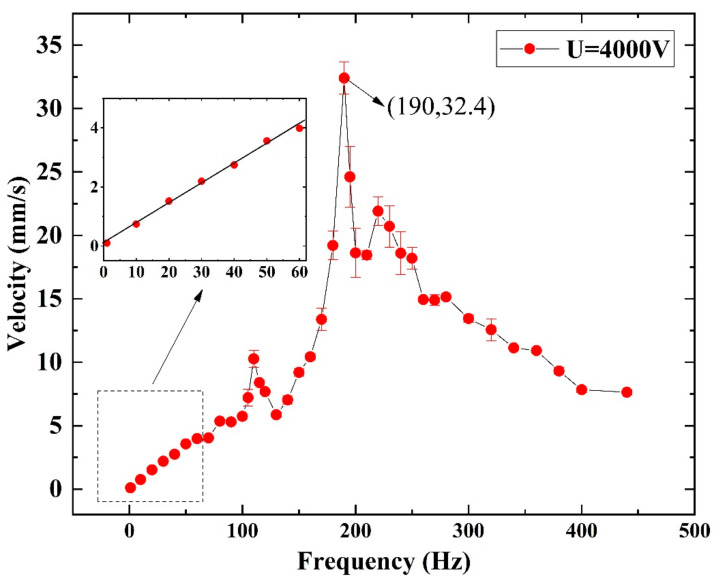
The horizontal crawling performance of the robot in the Φ18 acrylic pipe under Up = 4 kV (each experiment is tested five times and the results are averaged).

**Figure 11 micromachines-14-00618-f011:**
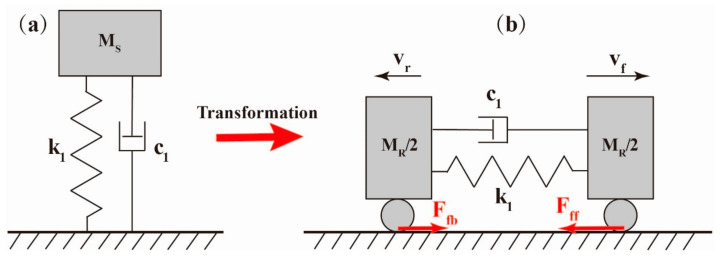
(**a**) The schematic simplification diagram of the SRDE (single-degree-of-freedom vibration system). (**b**) The schematic simplification diagram of the crawling robot (dual-degree-of-freedom vibration system similar to semidefinite system): the center SRDE is in a state of contraction to elongation, where *F_fb_* and *v_r_* are the friction and speed of the rear bristle, while *F_ff_* and *v_f_* are the friction and speed of the front bristle, respectively.

**Figure 12 micromachines-14-00618-f012:**
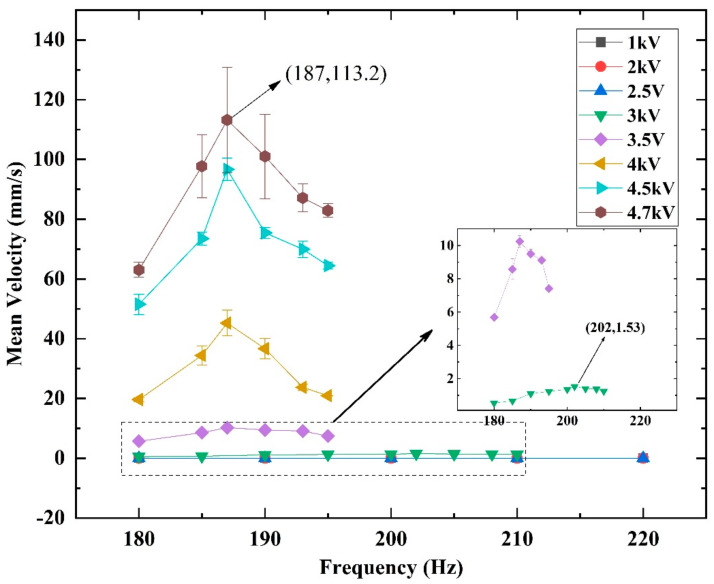
The crawling speed in Φ18 acrylic pipe under different peak voltages (*U_p_*).

**Figure 13 micromachines-14-00618-f013:**
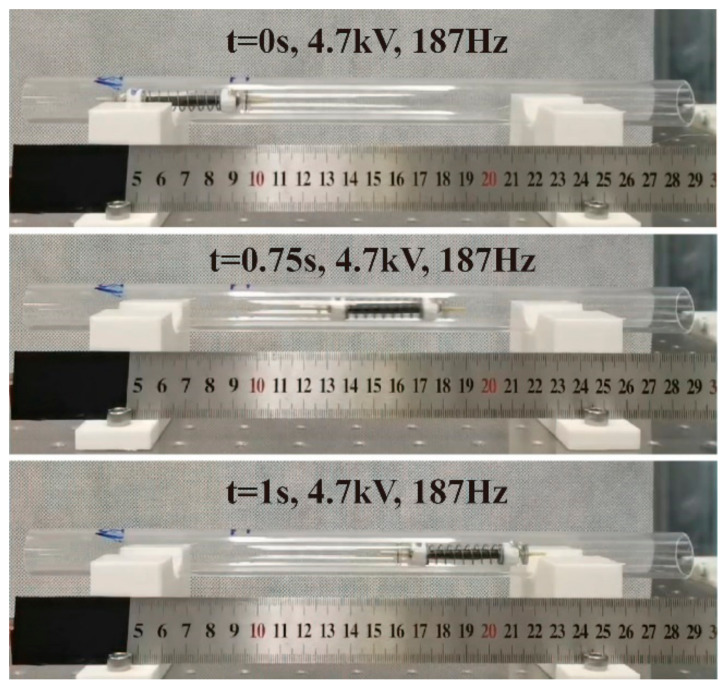
Photograph of the robot crawling in Φ18 acrylic pipe at *U_p_* of 4.7 kV and excited frequency of 187 Hz (Appendix A).

**Figure 14 micromachines-14-00618-f014:**
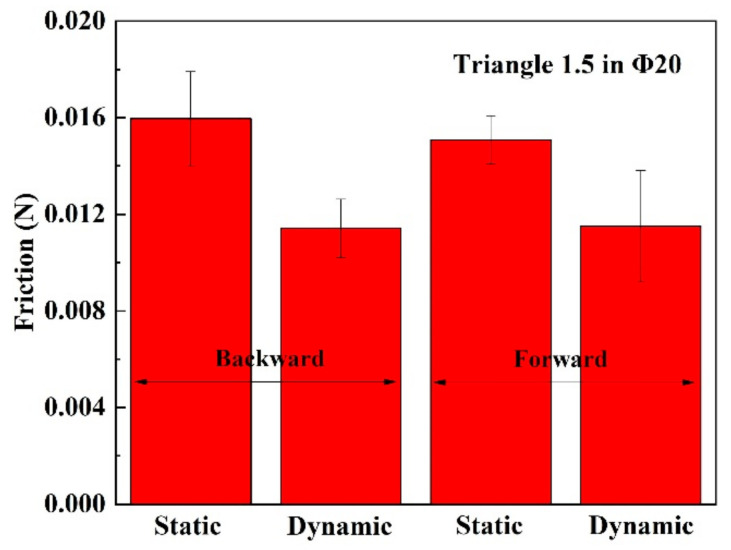
The friction test result of the robot with Triangle 1.5 bristles in Φ20 acrylic pipe.

**Figure 15 micromachines-14-00618-f015:**
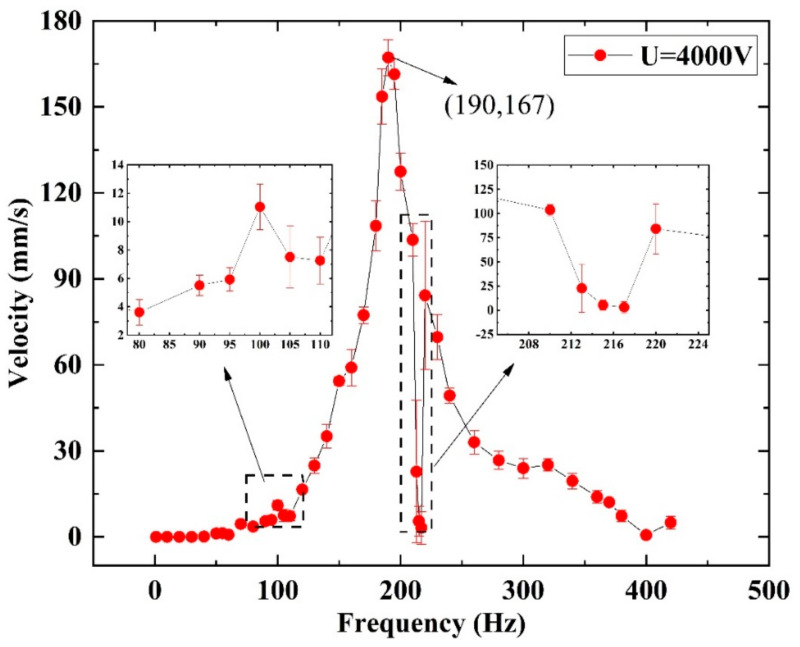
The hopping performance of the robot with Triangle 1.5 bristles in Φ20 acrylic pipe.

**Figure 16 micromachines-14-00618-f016:**
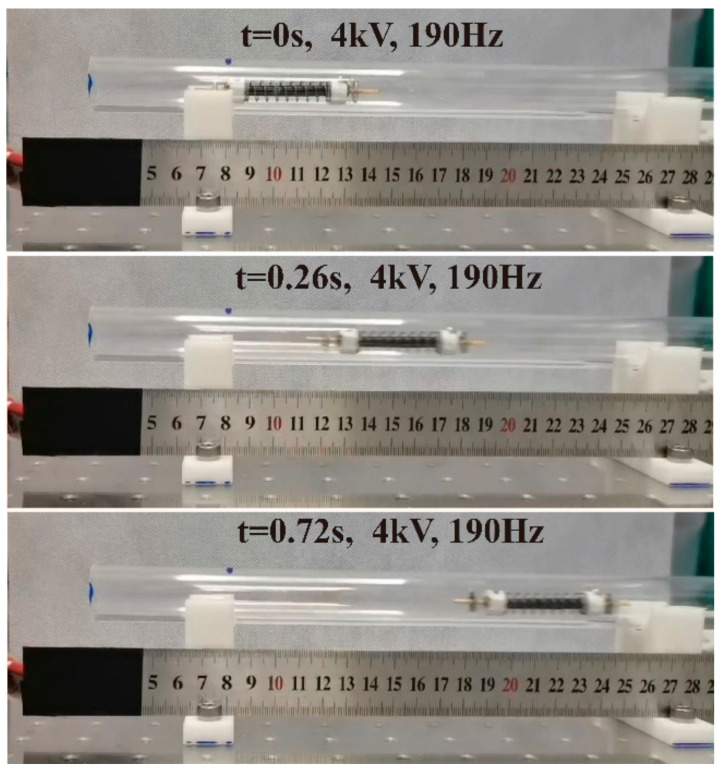
Photograph of the robot crawling in Φ20 acrylic pipe under *U_p_* of 4.0 kV and excited frequency of 190 Hz (Appendix A).

**Figure 17 micromachines-14-00618-f017:**
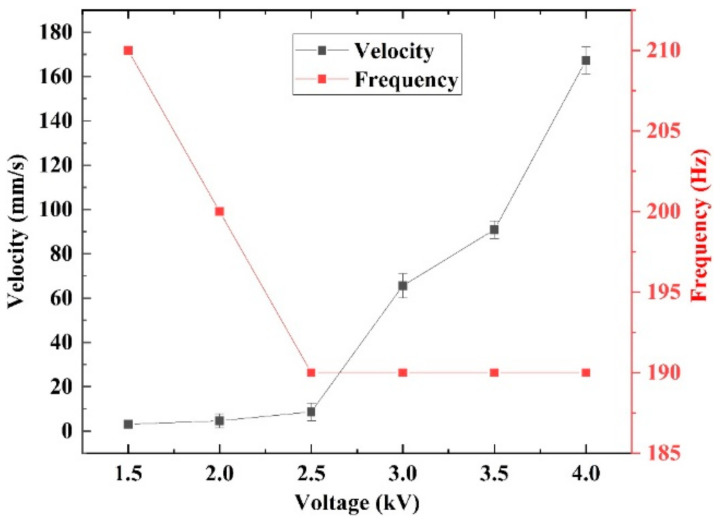
The hopping speed in Φ20 acrylic pipe under different peak voltages (*U_p_*).

**Figure 18 micromachines-14-00618-f018:**
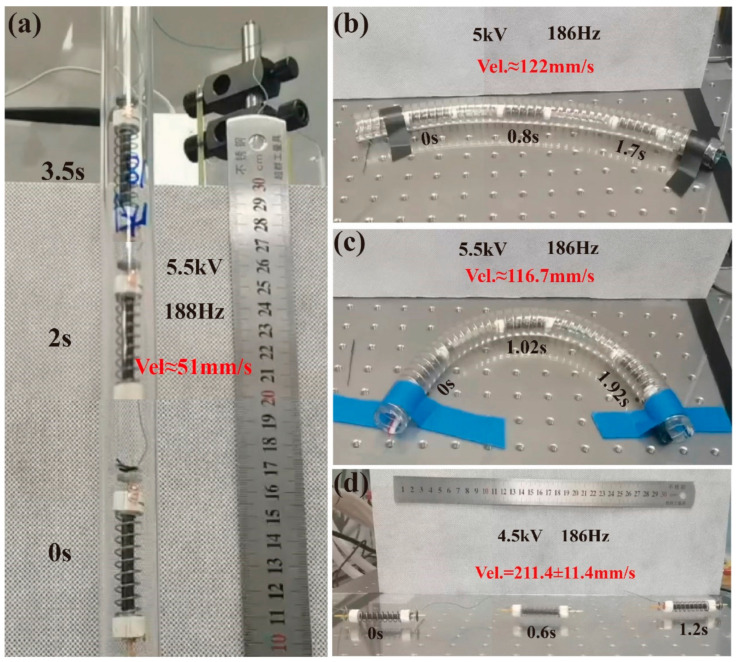
The crawling photographs of the robot in different environments. (**a**) Crawling in Φ18 acrylic pipe vertically (Appendix A). (**b**) Crawling in Φ20 PU spring tube with curving radius of 313.5 mm horizontally (Appendix A). (**c**) Crawling in Φ20 PU spring tube with curving radius of 110 mm horizontally (Appendix A). (**d**) Crawling on an acrylic plate horizontally (Appendix A).

**Table 1 micromachines-14-00618-t001:** Composite of the compliant electrode.

Composite	C Powder	RT625	Ecoflex 0030	Isopropanol
(wt. %)	3.1%	41%	6.2%	49.7%

## Data Availability

The data supporting reported results can be made available via requesting the corresponding author.

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
