# Peer review of "A Soft Robot Driven by a Spring-Rolling Dielectric Elastomer Actuator with Two Bristles"

_micromachines, 2023, doi:10.3390/mi14030618_

Round 1

Reviewer 1 Report

The authors fabricated a kind of soft robot which is actuated with dielectric elastomer. In general, the work is very interesting, and the paper is well-written. I think the paper can be published with minor revision. Here are some comments:

1 “Compared with other driving methods, i.e. shape memory alloy, piezoelectric ceramics, magnetic field, and pneumatic actuation, dielectric elastomers (DEs) in proposed robots have the comprehensive advantages of fast response, large strain, easy controlling and high efficiency of electromechanical conversion.”

It is true that the DEA can have advantages such as fast response, easy control, and so on. However, they are also the advantages of piezoelectric actuators. Moreover, the magnetic actuators are also fast response, and the strains are large. So please consider rephrasing it, and compare the DEAs and piezoelectric actuators more in details.

You can find some discussions about different actuators in the following literature:

“Soft Actuators for Soft Robotic Applications: A Review” https://doi.org/10.1002/aisy.202000128

“Bistable and Multistable Actuators for Soft Robots: Structures, Materials, and Functionalities” https://doi.org/10.1002/adma.202110384

“Flexible Actuators for Soft Robotics” https://doi.org/10.1002/aisy.201900077

2 The authors mentioned the stiffness of the spring they used is about 0.35 N/mm. Maybe you can very briefly talk about how the stiffness can influence the results.

3 Please provide the information about the high voltage amplifier (e.g., model, make). Please also provide the break down voltage of the dielectric polymer.

4 Figure 6,

(b) was not denoted in the figure. Moreover, the legend fonts are too small and the color lines are not clear enough.

5 Figure 8b,

The angles of the tips are not provided.

Reviewer 2 Report

Fig 6 (b) is the frequency sweep test but the x-axis is time instead of frequency.

What would be the best/optimum forward and backward friction ratio for motion speed? Does the material used for the bristle affect the performance? Also, does the geometry and shape of the bristle affect the performance?

Can the SRDE climb a tube that is inclined upwards? If yes, that is the max inclination? 

Does the SRDE rotates along its axis when it travel in a tube?

What is the minimum tube curvature the SRDE can operate?

Reviewer 3 Report

This manuscript presents the design of a tubular soft robot with a spring-rolled dielectric elastomer as well as two compliant bristles that help the robot adapt to various environments. The working principle of the robot under different scenarios and the fabrication process are first presented. The authors also investigated the dynamic and tribological aspects of the robot and demonstrated its crawling performance in pipelines of different geometries. 

Overall, this manuscript is clearly centered around a novelty which is the adaptivity of the soft robot to different landscapes, and it is recommended for publication after minor revision. Several comments to address are provided below. 

The authors are suggested to briefly explain what a rolling DE is early in the text, i.e. in section 2.1.

On page 3, line 92, the authors claim that the robot moves by dx3-dx1 in one cylce. Could this be verified experimentally later in the manuscript?

Page 5, Table 1: what does "et. al" mean? Shouldn't the weight percentage be given as numerical value?

Page 4, line 185: the authors claim that the max. amplitude reaches 1.72 mm, which is not in agreement with Fig. 7. Please explain. 

The authors are suggested to use the same scale on the displacement for all subfigures of Fig. 7.

There are several typographical or grammatical errors in the manuscript. Some examples include: 

* Page 1, line 34: "..., are attracted attention widely"

* Page 2, line 59: “... in the various environment.” 
